# Epidemiological characteristics of multidrug-resistant *Acinetobacter baumannii* ST369 in Anhui, China

Yi Huang,[1] Md Roushan Ali,[1] Wei Li,[1] Wanying Wang,[2] Yuanyuan Dai,[1] Huaiwei Lu,[1] Zhien He,[1] Yujie Li,[1] Baolin Sun[1]

**ABSTRACT** *Acinetobacter baumannii* is of growing concern for global health owing to its ability to evade the immune system and develop resistance to antibiotics. We analyzed *A. baumannii* using the National Center for Biotechnology Information (https://www.ncbi.nlm.nih.gov/pathogens) database and observed that the ST369 strain isolated in China constituted over 50% of the globally prevalent ST369 clones. This finding highlights the significant impact of ST369 on public health in China. In this study, we examined eight strains of multidrug-resistant *A. baumannii* ST369 isolated from a provincial hospital in China. We conducted whole-genome sequencing, comparative genomic analysis, and phenotypic experiments, and used the *Galleria mellonella* infection model to achieve a comprehensive understanding of the organism. In our study, we identified two mutations, G540A and G667D, on the *wzc* gene of *A. baumannii* ST369 that can potentially affect bacterial virulence and viscosity. We also verified the impact of these mutations on virulence and resistance. In addition, we examined two proteins, AB46_0125 and AB152_03903, which may play a role in virulence. These findings establish a foundation for future studies on clinical ST369 isolates harboring such mutations.

**IMPORTANCE** *Acinetobacter baumannii* is a major health threat due to its antibiotic resistance and ability to cause nosocomial infections. Epidemiological studies indicated that the majority of globally prevalent ST369 clones originated from China, indicating a significant impact on public health in the country. In this study, we conducted whole-genome sequencing, comparative genomics, and *Galleria mellonella* infection model on eight *A. baumannii* ST369 isolates collected from a provincial hospital in China to comprehensively understand the organism. We identified two mutations (G540A and G667D) on the *wzc* gene that can affect bacterial virulence and viscosity. We confirmed their impact on resistance and virulence. We also investigated the potential involvement of AB46_0125 and AB152_03903 proteins in virulence. This finding provides a theoretical reference for further research on *A. baumannii* ST369 clinical isolates with similar mutations.

**KEYWORDS** *Acinetobacter baumannii* ST369, *wzc*, multidrug-resistant *Acinetobacter baumannii*, whole-genome sequencing

*A*cinetobacter is an aerobic, non-motile, Gram-negative *Coccobacilli* that is catalase positive but non-glucose fermenting (1, 2). Among *Acinetobacter* species, *Acinetobacter baumannii* has recently emerged as a global concern owing to its rapid development of resistance to multiple antibiotics, despite being previously considered a lesser threat due to its lower infection rate than most other Gram-negative pathogens (3–7). *A. baumannii* is responsible for nosocomial infections, such as ventilator-associated pneumonia, bloodstream infection, skin and soft tissue infection, endocarditis, meningitis, and urinary tract infection. Patients admitted to intensive care units

Address correspondence to Zhien He, zhienhe@mail.ustc.edu.cn, Yujie Li, lyj2020@ustc.edu.cn, or Baolin Sun, sunb@ustc.edu.cn.

The authors declare no conflict of interest.

See the funding table on p. 12.

(ICUs) are particularly susceptible to these infections (8). Among various types of infection, bloodstream infection has the highest mortality rate, reaching 29%–63% (9, 10). *A. baumannii* bloodstream infections primarily affect intravascular and respiratory catheters, as well as surgical incisions, burn wounds, and the urethra. Notably, in 21%–70% of cases, the primary source of infection is unknown (11). Current research indicates that the incidence of community-acquired *A. baumannii* infection is increasing. Globally, approximately 45% of *A. baumannii* isolates that cause infection have been identified as multidrug-resistant (MDR) strains. In Latin America and the Middle East, the prevalence of infection by MDR strains is exceptionally high, reaching up to 70% (1, 3). Consequently, *A. baumannii* has been identified as a significant contributor to hospital-acquired infections on a global scale (12, 13). *A. baumannii* has also been declared as one of the most serious ESKAPE organisms by the World Health Organization, along with *Enterococcus faecium*, *Staphylococcus aureus*, *Klebsiella pneumoniae*, *Acinetobacter baumannii*, *Pseudomonas aeruginosa*, and *Enterobacter* spp. (14).

Currently, various virulence factors have been documented for *A. baumannii*, including outer membrane porin, capsular polysaccharides, phospholipases, proteases, and the iron chelation system (1, 15, 16). The capsular polysaccharide of *A. baumannii* is a significant virulence factor in this bacterium (2). Most *A. baumannii* carry a thick capsular polysaccharide that offers protection from external threats (17). The capsular polysaccharide of *A. baumannii* is reported to play a crucial role in bacterial defense against the host complement system (17, 18). The capsular polysaccharide (CPS) of *A. baumannii* is considered a key virulence factor because of its resistance to the complement system and decreased biofilm formation if absent. This reduces colonization ability and antibiotic resistance (18, 19). In addition, Geisinger and Isberg (20) have also shown that the CPS is involved in the antibiotic resistance of *A. baumannii*. They found that *A. baumannii* with mutant CPS was less resistant to some antibiotics, such as colistin and rifampicin; moreover, after antibiotic treatment, capsular polysaccharide product was increased (20). Although there are many studies on the CPS of *A. baumannii* (17–19, 21–24), the mechanism of CPS regulation in bacteria remains unclear.

*A. baumannii* has been found to survive for long periods on inert objects that lack nutrients, with studies indicating that it can persist for up to four months. In addition to *S. aureus* and *Pseudomonas*, *A. baumannii* is also frequently detected on the surface of inert medical instruments and in the hands of ICU medical staff (25). The strong colonization ability of *A. baumannii* allows it to exist and spread in both natural and medical environments, making it a persistent threat. It often colonizes medical equipment and devices, such as those for mechanical ventilation, sputum suction, and vascular access; these serve as important factors for the continued outbreak of *A. baumannii* in hospitals (26). The dissemination ability of *A. baumannii* has also led to some of its clones being identified worldwide, such as the international clones I–III that were initially identified in Europe (27, 28). Among them, some CC92° (Bartual scheme, also known as the Oxford scheme) clones belonging to *A. baumannii* international clone complex II, such as ST208 and ST195, are spread globally and have a high lethality rate (29–32).

We analyzed *A. baumannii* in the National Center for Biotechnology Information (NCBI) database and discovered that *A. baumannii* ST369, which was collected from China, constituted over 50% of the globally prevalent ST369 clones (33) (Table S1). It has been reported that, as an origin of bacteremia, the frequency of pneumonia was higher in ST369 than in non-ST369 (34). Therefore, we investigated the prevalence of *A. baumannii* ST369 in Anhui Province, China, and analyzed the virulence and antibiotic resistance of certain clinical strains. Furthermore, we identified the position mutation on the *wzc* gene, which is associated with the stickiness of *A. baumannii*, thereby providing a foundation for further research.

**TABLE 1** Basic clinical information of eight MDR *Acinetobacter baumannii* isolates[a]

| Isolate | Age | Underlying disease | Sample collected | Ward |
|---|---|---|---|---|
| AB47 | 68/F | Respiratory failure | 2021/11/17 | Intensive medicine |
| AB46 | | | 2021/11/18 | ward |
| AB58 | | | 2021/11/18 | |
| AB60 | | | 2021/11/26 | |
| AB59 | | | 2021/11/28 | |
| AB145 | 56/F | Respiratory-related | 2022/3/4 | Respiratory and critical |
| AB144 | | diseases | 2022/3/4 | care ward |
| AB152 | | | 2022/3/7 | |

[a]All strains were recovered from sputum.

## MATERIALS AND METHODS

### Bacterial strains and growth conditions

We obtained eight clinical strains of MDR *Acinetobacter baumannii* from two patients at the Anhui Provincial Hospital in China between December 2021 and March 2022 (Table 1). Bacterial colonies were isolated by inoculating the isolates on blood plates and incubated at 37°C for 24 h. The VITEK2 Compact system (bioMérieux, France) was then used to identify positive strains and perform the antibiotic sensitivity test. All isolates were preserved in 40% (vol/vol) glycerol broth at −80°C until further use.

### Construction of plasmids and strains

The kanamycin resistance gene on the pET28a plasmid was obtained using pUC19-Hind III-F and pUC19-EcoR I-R. Plasmids and primers used in this experiment are listed in Table 2.

Then, the pUC19 plasmid was cut with Hind III and EcoR I, and the fragment was ligated with the plasmid in the presence of T4 ligase to form the pUCk19 plasmid.

The *wzc* gene on the bacterial genome was obtained using *wzc*-BamH I-F and *wzc*-Sal I-R (Table 2). The pUCk19 plasmid was cut with BamH I and Sal I, and the fragment was ligated with the plasmid in the presence of T4 ligase to form the pUCk*wzc* plasmid. Next, the recombinant plasmid was transformed into competent *Escherichia coli* cells by heat excitation and stored at −80°C until use. In addition, the recombinant plasmid was transformed into *A. baumannii* competent cells by electric shock and stored at −80°C until use.

### Growth curves

In this study, we determined the growth curve of *A. baumannii* in the Luria-Bertani (LB) medium. Cultures were grown overnight, diluted to an $OD_{600}$ of 0.05, and grown

**TABLE 2** Plasmids and oligonucleotide primers

| Plasmid or primer | Description or oligonucleotide (5′–3′) | Source or application |
|---|---|---|
| Plasmid | | |
| pUCk19 | Transformed by inserting a kanamycin resistance gene into the pUC19 plasmid | Modified by our lab and used for this experiment |
| Primers | | |
| pUCk19-F | CAGGAAACAGCTATGAC | Verify whether the plasmid is inserted |
| pUCk19-R | TGTAAAACGACGGCCAGT | |
| *wzc*-F | CAGTGGAAACTCATTGCTCT | Verify whether the recombinant plasmid is correct |
| *wzc*-R | GCGCTAGCACGTTGAATAT | |
| pUC19-Hind III-F | AAGCTTTGCATGCCTGCAGGTCGA | Construction for pUCk19 plasmid |
| pUC19-EcoR I-R | GAATTCATGAGCCATATTCAACGG | |
| *wzc*-BamH I-F | CGCGGATCCAACCGGATCATTTGATCCG | Construction for pUCk*wzc* plasmid |
| *wzc*-Sal I-R | CGCGTCGACGCATTGATATGCAGCCTCATA | |

in 96-well plates under the following conditions: temperature, 37°C; rotation speed, 200 rpm; and shaking. The absorbance of the culture solution at 600 nm was measured every 0.5 h until the peak value was reached and remained constant.

## Mucoviscosity assay

*A. baumannii* viscosity was determined using the string test (35). Strains that form filaments when stretched with a sterile loop or the tip of a pipette are considered more viscous. *A. baumannii* was cultured overnight at 37°C in LB medium or LB Kan medium at 200 rpm. On the next day, the culture was diluted to an $OD_{600}$ of 1 and centrifuged at $2,000 \times g$ for 5 min, and the $OD_{600}$ of the supernatant was measured every single minute.

## *Galleria mellonella* infection model

The virulence of *A. baumannii* isolates was evaluated using a *Galleria mellonella* infection model. Larvae, weighing 0.2–0.3 g, were kept in the dark and used within three days of shipment (Keyun Bio). Prior to injection, the bacterial pellet was washed with sterile saline or sterile saline with kanamycin and diluted to a concentration of $1 \times 10^8$ CFU/mL. Then, using a 1-mL insulin syringe (Shanghai Kindly Ent Dev), 10 µL bacterial suspension was injected into the center of each larva's second abdominal cavity. Ten larvae were randomly selected for injection. Each treatment was performed in triplicate for a total of 30 larvae. After injection, the larvae were incubated at 37°C, and survival was monitored every 12 h for 3 days. Death was considered to have occurred when larvae were no longer responsive to touch. Larvae that were not injected or were injected with 10 µL of sterile saline were used as negative controls.

## Whole-genome sequencing, assembly, and annotation

In total, all isolates were sequenced—six by second-generation sequencing and two by third-generation sequencing. Whole-genome sequencing of *A. baumannii* was performed using the PacBio RS II and Illumina HiSeq 4000 platforms at the Nuosai Jiyin Zu Research Center Limited Company, Beijing, China. Four SMRT cellular zero-mode waveguide arrays for sequencing were used with the PacBio platform to generate subhead sets. PacBio subheads (<1 kb in length) were removed. The pbdagcon program was used for self-correction (https://github.com/PacificBiosciences/pbdagcon). Draft genomes were uncontroversial fragment sets that were assembled using a Celera Assembler against high-quality corrected circular consensus sequence subhead sets. To improve the accuracy of genome sequencing, GATK (https://www.broadinstitute.org/gatk/) and the SOAP toolkit (SOAP2, SOAPsnp, and SOAPindel) were used for single-base correction. A new hybrid assembly, comprising short Illumina reads and long PacBio reads, was performed using Unicycler v0.4.8 (36) and annotated using the rapid prokaryotic genome annotation tool Prokka 1.14.6 (37). The plasmid map was drawn using BRIG 0.95 and Easyfig 2.2.5 (38, 39).

## Genome profiling and comparative genomic analysis

Acquired antimicrobial resistance genes [ARG (https://github.com/tseemann/abricate)] were identified using ABRicate version 1.0.1 by aligning genome sequences from the ResFinder and NCBI databases (40). The virulence factors of the isolates were identified by matching the genome sequences with the Virulence Factor Database (VFDB) using Kleborate and ABRicate (40, 41). Multilocus sequence typing (MLST) was performed using MLST 2.1 (https://cge.food.dtu.dk/services/MLST/) (42). Comparative genomic and phylogenetic analyses were performed on different isolates using the HarvestTools toolkit (Parsnp, Gingr, and HarvestTools) and BacWGSTdb. Phylogenetic trees based on single-nucleotide polymorphisms (SNPs) were constructed for all isolates using the maximum likelihood method. Interactive Tree of Life (iTOL) v5 (http://itol.embl.de/) was

used to illustrate phylogenetic trees (43–45). SNPs were detected in 21,072,329 complete genomes using Snippy (https://github.com/tseemann/snippy).

## Statistical analyses

All analyses were performed using Prism software (GraphPad Software, La Jolla, CA, USA) and RStudio. Error bars represent the SEM. All experiments were repeated at least three times.

## RESULTS

### Global distribution of *A. baumannii* ST369

We collected eight clinical *A. baumannii* isolates from a tertiary hospital in Anhui Province, China. Through MLST identification and genome sequencing, we determined that the isolates were ST369 clones. Further research using the NCBI database revealed that there are approximately 142 ST369 clones as of 2023, most prevalent in the United States and China (Fig. 1; Table S1). According to Fig. 1A, ST369 is predominantly identified in countries associated with high human traffic, such as China, India, the United States, Mexico, Germany, and France. These findings suggest that the *A. baumannii* strain ST369

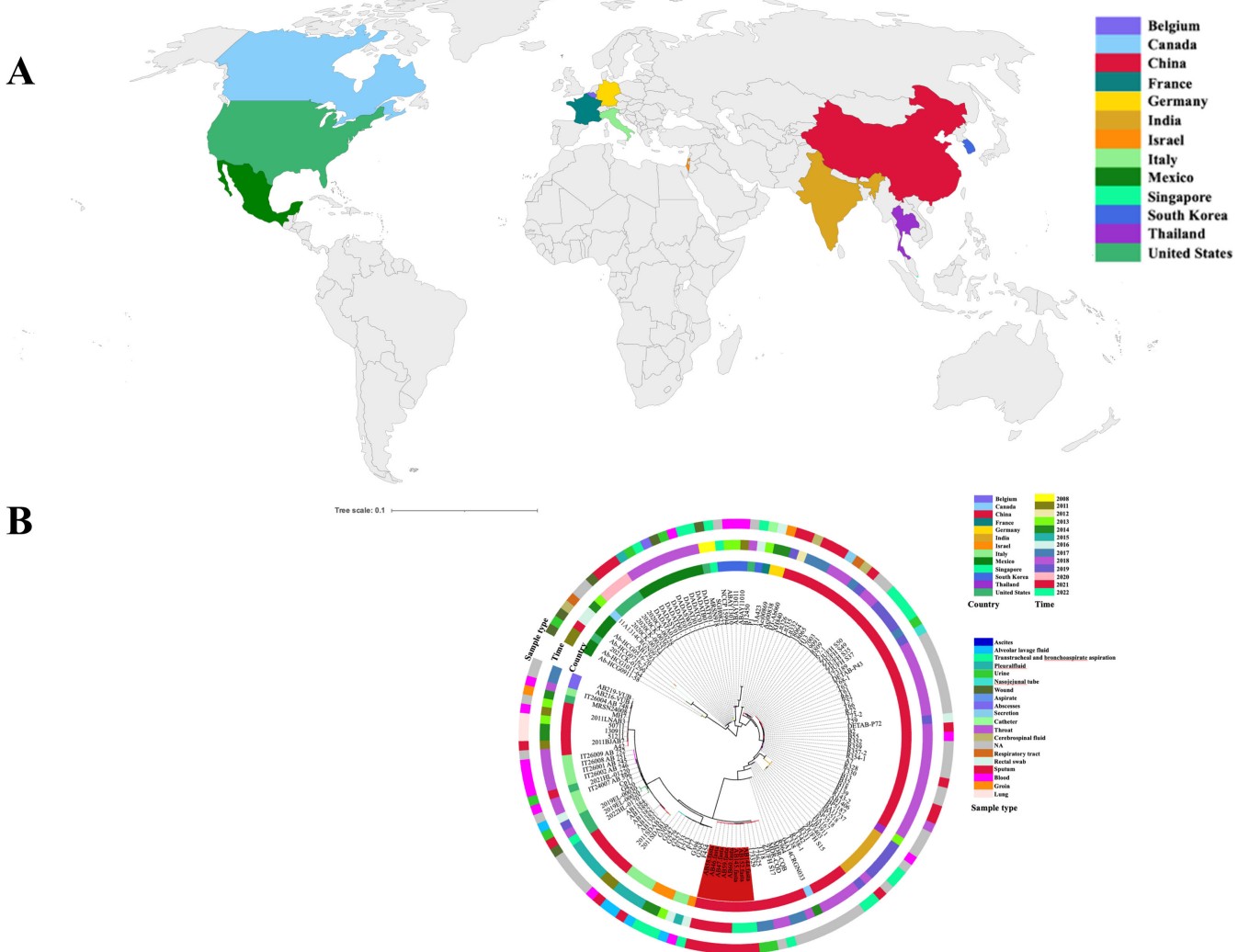

FIG 1 Global distribution of *A. baumannii* ST369. (A) Distribution of ST369 in different countries. (B) Phylogenetic tree analysis of the collected isolates with other ST369 worldwide. The red font in the inner ring represents the eight isolates collected in this study.

is a globally prevalent clone and that the isolates collected in this study are evolutionarily similar to those found in China (Fig. 1B), indicating that *A. baumannii* ST369 is prevalent in China.

## *A. baumannii* ST369 exhibits multidrug resistance characteristics

In this study, we found that all isolates belonged to *A. baumannii* ST369 and were resistant to multiple antibiotics. The isolates showed resistance to seven different antimicrobial classes (46), including the antimicrobial compounds piperacillin/tazobactam, ticarcillin/clavulanic acid, ceftazidime, cefepime, imipenem, meropenem, tobramycin, cefoperazone/sulbactam, ciprofloxacin, levofloxacin, and co-trimoxazole. The only sensitivity was observed to colistin, but resistance to doxycycline was found (Fig. 2). Based on the antibiotic resistance exhibited by those isolates (Fig. 2B), there exists the *bla* gene that can confer resistance to β-lactam antibiotics (Fig. 3A).

In addition, the isolates showed low resistance to tetracyclines, including minocycline and tigecycline, indicating the absence of certain tetracycline resistance genes (Fig. 2A and 3A). These isolates are solely resistant to polymyxins, which are known to be highly toxic to human kidneys and serve as the last defense against Gram-negative bacteria. Consequently, these isolates possess significant research value, which is helpful for the investigation of the toxicity mechanisms of MDR strains.

## AB46, AB58, AB59, and AB152 are highly virulent with strong adhesion characteristics

To assess the viscosities and virulence of these isolates, we first inoculated the isolates on fresh sheep blood medium and observed the colony morphology. It is clearly shown

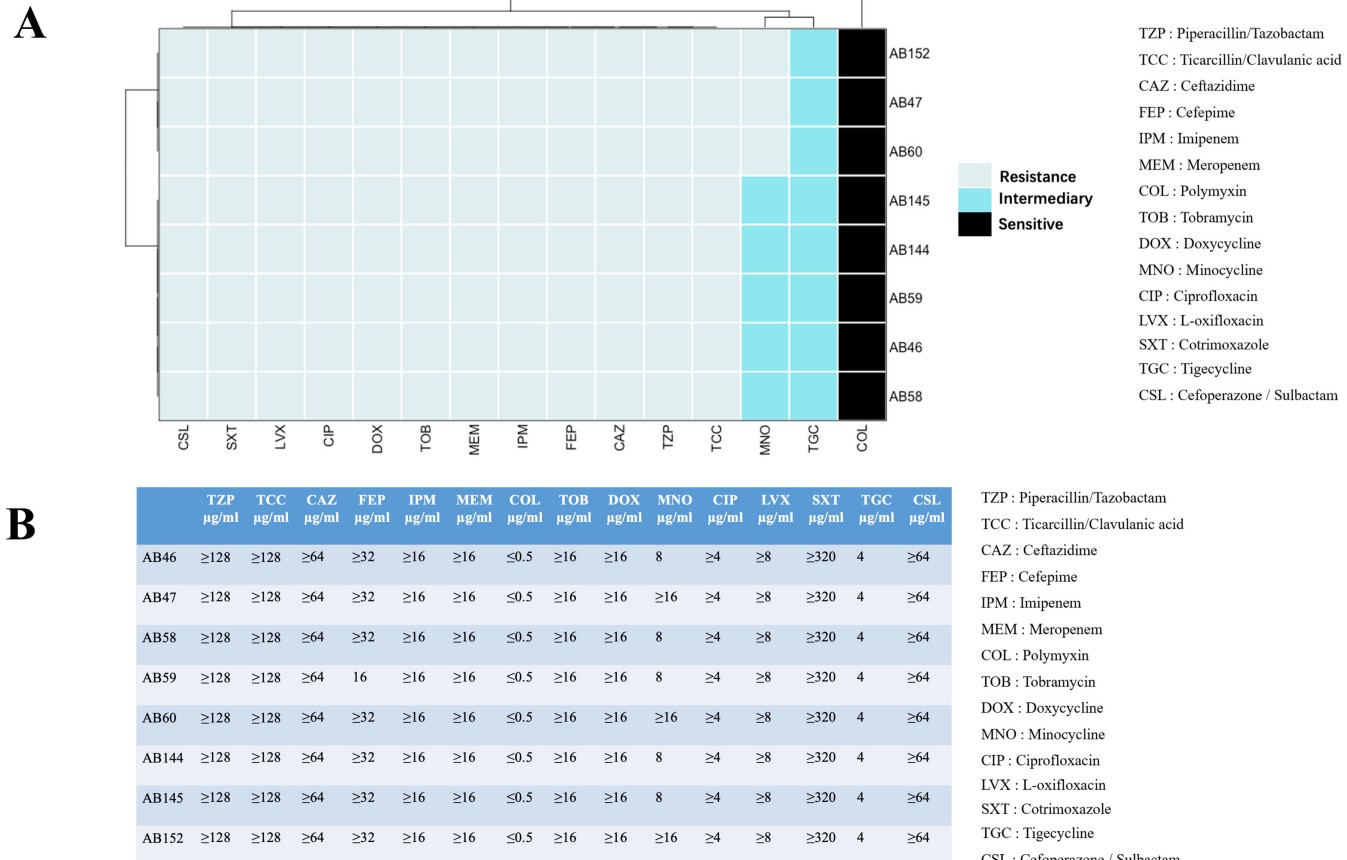

**FIG 2** Resistance of isolates. (A) Resistance of isolates to different types of antibiotics. (B) MIC of various antibiotics for the isolated clones.

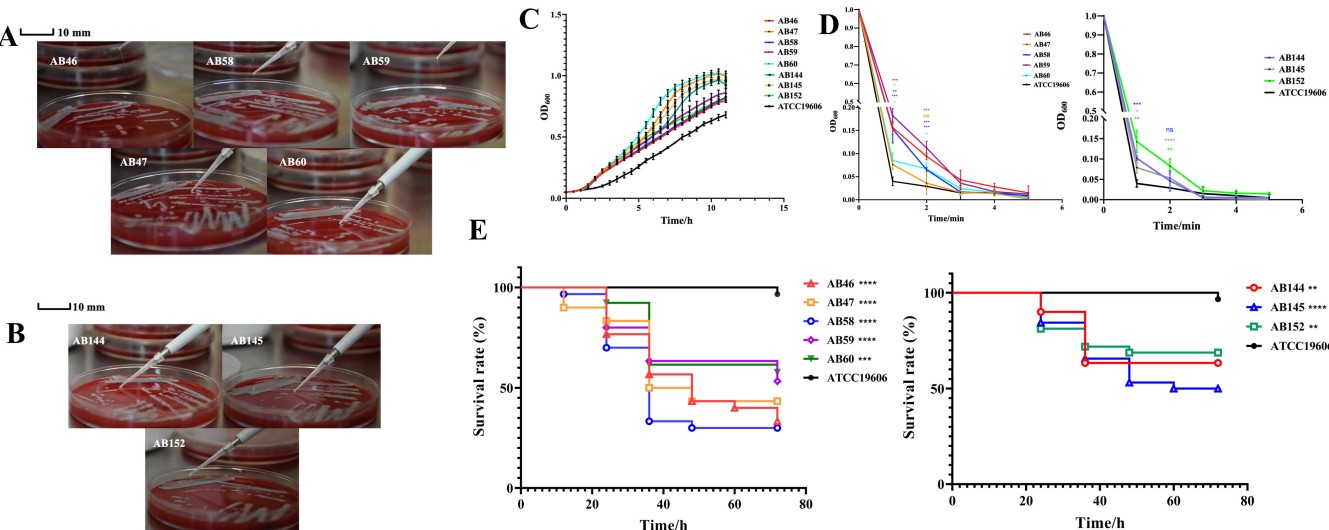

**FIG 3** Virulence and viscosity models of isolates ($n$ = 3). (A) String lengths of AB46, AB47, AB58, AB59, and AB60. (B) String lengths of AB144, AB145, and AB152. (C) Growth curves of isolates. (D) Viscosity of the isolates. (E) *Galleria mellonella* models of isolates in 72 hr; no dead larvae were observed in the negative controls. All clones were compared with ATCC19606. *, $P < 0.05$; **, $P < 0.01$; ***, $P < 0.001$; ****, $P < 0.0001$; ns, no significance.

that most of the single colonies of AB46, AB58, AB59, and AB152 stick together to form plaques, whereas the single colonies of AB47, AB60, AB144, and AB145 are scattered (Fig. S1). From this, we can preliminarily see that the viscosity of AB46, AB58, AB59, and AB152 is stronger than that of AB47, AB60, AB144, and AB145. Next, to evaluate the viscosity and virulence of these isolates, we used ATCC19606 (*A. baumannii* global strain clonal complexes II) as a control to detect the growth curve of the isolate, the *G. mellonella* infection, and the viscosity of the bacterial liquid (47). It can be seen from Fig. 2A and B that AB46, AB58, and AB59 can form strings of more than 10 mm (Fig. 3A), whereas AB152 has a weaker ability to form strings of less than 10 mm (Fig. 3B). In addition, none of the other isolates could form a string. This can indicate that AB46, AB58, AB59, and AB152 are more viscous than other isolates, which was also hinted at this point from the side by the growth curves of those isolates. Generally, more viscous strains will grow slightly slower; it is clear that the growth rates of AB46, AB58, AB59, and AB152 are almost similar and are slower than those of AB47, AB60, AB144, and AB145 (Fig. 3C). Next, we adjusted the initial $OD_{600}$ of the bacterial culture to 1, centrifuged at $2{,}000 \times g$ for 5 minutes, and measured the $OD_{600}$ of the supernatant every minute. The supernatants of AB46, AB58, and AB59 were significantly more turbid than those of AB47 and AB60 at 2–3 minutes. The same was applicable to AB152 (Fig. 3D). These findings further proved that AB46, AB58, AB59, and AB152 belonged to the isolates with strong stickiness.

Highly viscous *K. pneumoniae* strains are also reported to have strong toxicity (48, 49). However, no study has linked the high viscosity of *A. baumannii* to its virulence. To explore whether *A. baumannii* had the same properties, we used the *Galleria mellonella* infection model to detect the virulence of these isolates (Fig. 3E). After 24 h, it is clear that AB46, AB58, and AB59 resulted in a higher number of *G. mellonella* deaths than ATCC19606 and other isolates, although AB152 did not show stronger virulence than other isolates. This indicated that AB46, AB58, and AB59 were the more virulent of the isolates, and AB152 was the less virulent isolate. These results suggest that high viscosity in *A. baumannii* is not directly related to stronger virulence but can partially reflect the level of virulence.

## Phylogenetic analysis of *A. baumannii* ST369

To investigate the mechanisms behind the high virulence of the isolates and the resistance, we performed whole-genome sequencing. In the eight isolates, 39 virulence

genes and 15 resistance genes were found (Fig. 4A; Tables S2 and S3). The most commonly detected virulence genes were *pga*, *csu*, and *bas*. Both *pga* and *csu* are associated with biofilm formation, whereas the *bas* gene cluster is linked to bacterial efflux pumps (50). Several ESBL (extended-spectrum β-lactamases) genes were detected, including $bla_{TEM-12}$, $bla_{OXA-23}$, $bla_{ADC-30}$, and $bla_{OXA-66}$; these genes cause resistance to β-lactam antibiotics. Additionally, these isolates exhibited resistance to tetracycline and macrocyclic peptide-based antibiotics through *tetB* and *msrE* genes (51). We conducted a phylogenetic analysis of those isolates and identified AB46, AB47, AB58, AB59, and AB60 as belonging to the same strain, whereas AB144, AB145, and AB152 belonged to another strain of *A. baumannii* (Fig. 4B). Although the phylogenetic tree indicated that the bacteria isolated from the same patient were not the same species of *A. baumannii*, SNP analysis revealed that the differences in SNP among these bacteria were minimal and could be considered the same species (Fig. S2). For third-generation genome sequencing and plasmid analysis, we selected the AB46 and AB152 strains, which are known for their high virulence and stickiness, respectively. $bla_{OXA-23}$ was found only in the AB46 plasmid; The plasmids of AB46 and AB152, which were highly viscous isolates, contained only a limited number of identified virulence and resistance genes, suggesting that these genes are integrated into the genome. There is a high degree of similarity in the virulence genes of the isolates, although there are varying levels of virulence and stickiness. This similarity can be attributed to point mutations in certain virulence genes.

## SNP analysis of *wzc* from ST369 isolates

AB46 and AB152 were selected as the reference genomes owing to their strong viscosity and virulence. Using these as reference genomes, we aimed to identify site mutations in the *wzc* gene from several other isolates. Our findings indicate that AB58 and AB59 have a similar *wzc* to AB46, resulting in similar viscosity and virulence among the three isolates. However, AB47 and AB60 contain a missense mutation in *wzc*. Additionally, AB60 contains a missense mutation in an unidentified gene (*AB46_0125*) (Fig. 5A). From Fig.

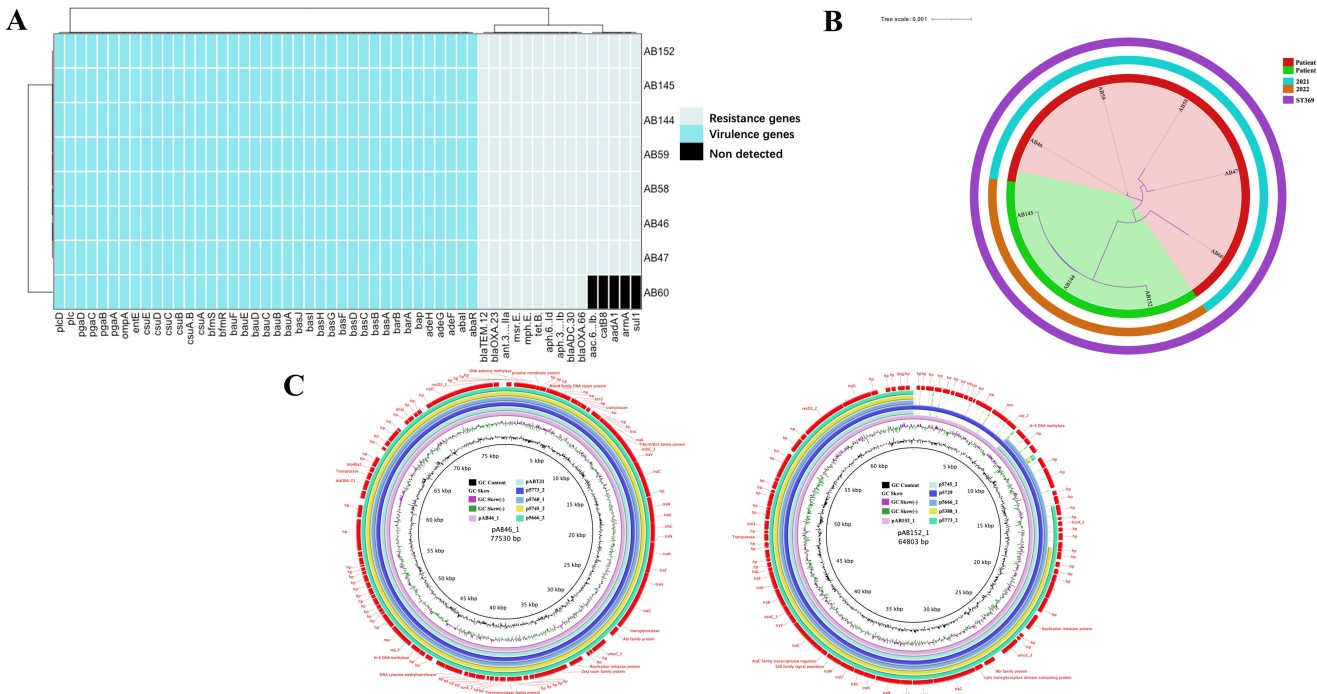

**FIG 4** Phylogenetic analysis of ST369 clones. (A) Virulence and resistance genes of isolates. (B) Phylogenetic tree of isolates. (C) Plasmids of AB46 and AB152.

5B, it can be seen that AB144 and AB145 have missense mutations in *wzc*. Additionally, AB144 and AB145 have both consensus and missense mutations in an unknown gene (*AB152_03903*) (Fig. 5B). Further analysis revealed that AB47 and AB60 had mutations from guanine (G) to cytosine (C) at position 1619 of *wzc*, resulting in a change from Gly to Ala at the 540th encoded amino acid in contrast to AB46. Similarly, AB144 and AB152 contained a mutation from guanine (G) to adenine (A) at position 2,000 of *wzc*, resulting in a change from Gly to Asp at the 667th amino acid in contrast to AB152 (Fig. 5B and C). Although AB47 and AB60 have the same mutation in *wzc*, we observed that AB47 exhibited significantly stronger toxicity than AB60, which suggests that the relationship between the virulence and viscosity of *A. baumannii* is not as closely linked as that of *K. pneumoniae*. Additionally, the difference in virulence between the two strains could be attributed to the *AB46_0125* gene. However, the function of AB46_0125 remains unknown and requires further investigation, particularly regarding its potential impact on the virulence of *A. baumannii*. Overall, we speculate that mutations of *wzc* may influence the virulence and viscosity of *A. baumannii*.

## The *wzc* mutation can affect the viscosity and virulence of *A. baumannii* ST369

Our prior investigations revealed that the *wzc* gene from *K. pneumoniae* contributes to CPS production, which affects the virulence of the bacteria (52). Based on our analyses, we concluded that *wzc* mutations can enhance the viscosity and virulence production.

To investigate the potential impact of mutations on AB46, AB58, AB59, and AB152 on bacterial virulence and viscosity, we constructed pUCk*wzc* plasmids incorporating the *wzc* genes from AB60, AB46, and AB152. These plasmids were subsequently transformed into bacterial cells for further analysis. In this study, we acquired three strains: AB60-*wzc*, AB60-*wzc*-A2, and AB60-*wzc*-A3; these strains were transformed with the *wzc* genes from AB60, AB46, and AB152, respectively. Subsequently, the three newly obtained strains were utilized for colony viscosity detection, bacterial liquid viscosity detection, and *G. mellonella* infection experiments. Based on the colony morphology and string length, it is evident that the strains transformed with the *wzc* gene of AB46 and AB152 can form strings, resulting in bacterial cultures with increased viscosity (Fig. 6A and B; Fig. S4). In

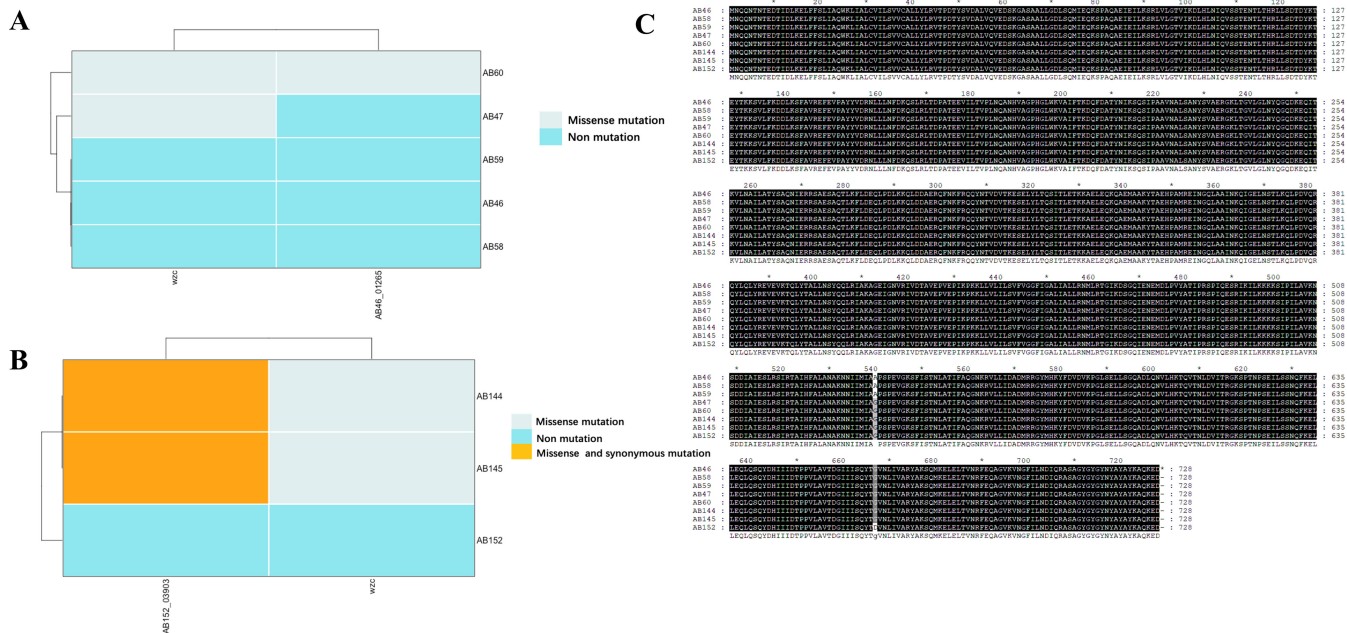

**FIG 5** Analysis of *wzc* mutation in *A. baumannii* ST369. (A) *wzc* mutation in AB46, AB47, AB58, AB59, and AB60. (B) *wzc* mutation in AB144, AB145, and AB152. (C) Comparison of Wzc between these isolates.

addition, both AB60-*wzc*-A2 and AB60-*wzc*-A3 exhibited higher lethality than AB60-*wzc*. Moreover, AB60-*wzc*-A2 demonstrated significantly higher toxicity than AB60-*wzc*-A3 (Fig. 6C), which agrees with prior results (Fig. 3). The results suggest that mutations at the *wzc* site affect the stickiness and toxicity of bacteria. A mutation at amino acid 540 of *wzc* has a greater impact than at amino acid 667. It also suggests that the mutation from purine to pyrimidine has a greater impact than mutations between purines (Fig. 5C). In brief, cysteine mutation in *wzc* affects bacterial virulence and antibiotic resistance. Further investigation is required to understand the mechanism behind this.

## DISCUSSION

Recent studies have indicated that ST369 is associated with a higher risk of pneumonia and bacteremia (34). Additionally, it has been reported that patients with ST369-associated bacteremia had a higher incidence of leukopenia than patients without ST369 (34). The CPS of *A. baumannii* is the main virulence factor, which is crucial for its survival both *in vivo* and *in vitro* (17). It has also been reported that CPS is associated with the onset of bacteremia (53).

Our study revealed that ST369 is a highly prevalent monoclonal strain of *A. baumannii* in China (Fig. 1; Table S1). Although the isolates we collected had highly similar virulence genes and antibiotic resistance genes, there were variations in their resistance and virulence phenotypes. AB60 lacks several resistance genes; however, it exhibits significant multidrug resistance similar to other isolates (Fig. 2A and 4A). This may be attributed to multiple resistance genes conferring the same resistance in AB60 for a particular antibiotic, allowing it to maintain resistance even when one or more genes are missing. Although all isolates possess the same virulence genes, AB60, AB144, AB145,

**A** 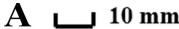 **10 mm**

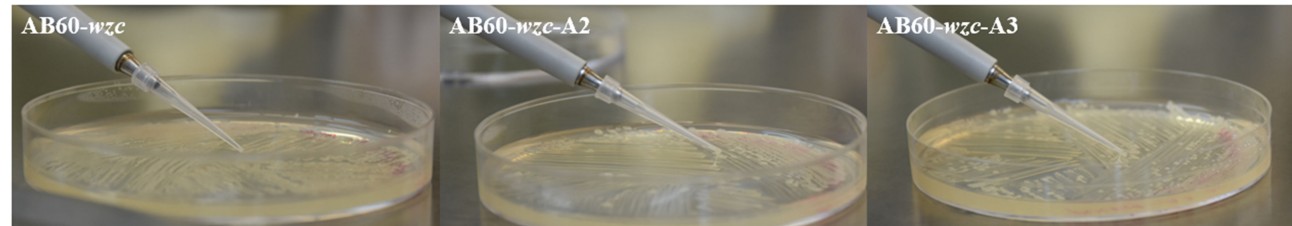

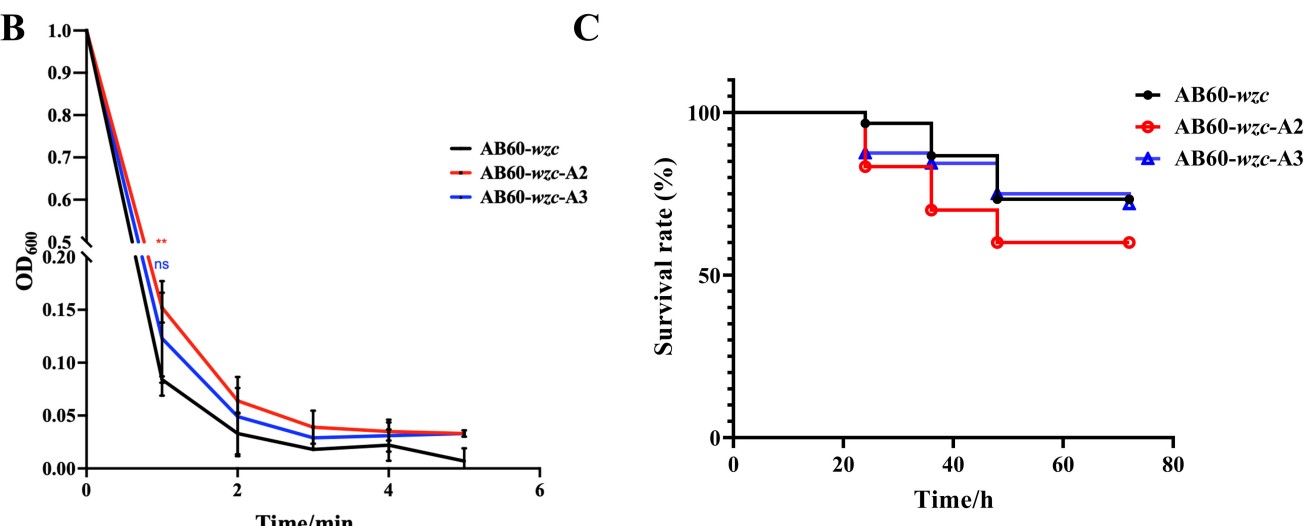

FIG 6 Functional validation of the *wzc* mutation site. (A) String lengths of AB60-*wzc*, AB60-*wzc*-A2, and AB60-*wzc*-A3. (B) Viscosity of AB60-*wzc*, AB60-*wzc*-A2, and AB60-*wzc*-A3. (C) *Galleria mellonella* models of AB60-*wzc*, AB60-*wzc*-A2, and AB60-*wzc*-A3 in 72 hr, no dead larvae were observed in the negative controls.**, $P < 0.01$; ns, no significance.

and others exhibit a notable decrease in both virulence and viscosity. In contrast, only AB47 shows a significant reduction in viscosity, with a lesser degree of virulence reduction (Fig. 3E and 4A). A comparative analysis of SNP revealed that AB47 and AB60 had mutations in wzc, which is related to CPS production. This may be the main reason for their reduced viscosity, unlike AB46. The same reason was applicable for AB144 and AB145 (Fig. 3C, D, and 5A B, ). However, the change in AB47 virulence is inconsistent with the viscosity change. AB60 differs from AB47 not only in the *wzc* mutation site but also in an unknown gene mutation (*AB46_0125*). Thus, the reduced toxicity of AB60 is likely due to this mutant protein, and we observed a similar phenomenon in AB144 and AB145 (*AB152_03903*). Further research is needed to elucidate the functions of these two unknown proteins and the impact of point mutations on their activity. We conducted experiments to determine the effect of a point mutation in the *wzc* gene on the stickiness of the isolates. Our results showed that introducing the mutated *wzc* gene in AB60 increased the virulence and stickiness of the bacteria, whereas introducing the *wzc* gene into AB46 resulted in even stronger virulence and stickiness compared with the bacteria with the introduced *wzc* gene in AB152. These findings suggest that the mutation of the 540th amino acid in Wzc has a greater impact on protein function than the mutation in the 667th position (Fig. 5C). Further investigation is required to understand the mechanism underlying this mutation-induced change.

In conclusion, we conducted an epidemiological analysis of globally prevalent *A. baumannii* ST369 and found that clones isolated in this study were more prevalent in China, indicating the prevalence of *A. baumannii* in China. We then conducted whole-genome sequencing of the isolates of *Acinetobacter baumannii* ST369 that were collected from the Anhui Provincial Hospital in China. Finally, we analyzed these strains' virulence and resistance and confirmed the impact of the *wzc* site mutation on the viscosity and virulence of *A. baumannii*. Overall, the prevalence of MDR hypervirulent *A. baumannii* is on the rise; therefore, it is urgent to pay more attention to limit further spread.

## ACKNOWLEDGMENTS

Thanks to Dr. Xueqin Shu and Ms. Yu Cheng from the Sun Baolin Laboratory, University of Science and Technology of China, who helped me during my experiments.

This research was supported by grants from the National Natural Science Foundation of China (32070132), the Fundamental Research Funds for the Central Universities (YD9100002013), the National Key Research and Development Program of China (2021YFC2300300), and the Strategic Priority Research Program of the Chinese Academy of Sciences (XDB29020000).

The experiment was designed by Zhien He and Yi Huang and performed by Yi Huang, Zhien He, Md. Roushan Ali, and Yujie Li. Isolates were collected by Zhien He, Yi Huang, Yuanyuan Dai, and Huaiwei Lu. The ATCC19606 clone was obtained from Wanying Wang; the pUCk19 plasmid was designed by Wei Li; the draft manuscript was written by Yi Huang and critically revised by Zhien He and Md Roushan Ali. All authors have read and approved the final manuscript.

## AUTHOR AFFILIATIONS

[1]Department of Oncology, The First Affiliated Hospital of University of Science and Technology of China, Division of Life Sciences and Medicine, University of Science and Technology of China, Hefei, China

[2]Intensive Care Unit, Biomedical Research Center, Shenzhen Institute of Translational Medicine, Health Science Center, The First Affiliated Hospital of Shenzhen University, Shenzhen Second People's Hospital, Shenzhen, China

## AUTHOR ORCIDs

Md Roushan Ali  http://orcid.org/0000-0002-8894-2566
Zhien He  http://orcid.org/0000-0003-4679-7839

Baolin Sun ⓘ http://orcid.org/0000-0002-8209-1246

## FUNDING

| Funder | Grant(s) | Author(s) |
| --- | --- | --- |
| MOST | National Natural Science Foundation of China (NSFC) | 32070132 | Baolin Sun |
| Foundation for Fundamental Research of China of the central university | YD9100002013 | Baolin Sun |
| MOST | National Key Research and Development Program of China (NKPs) | 2021YFC2300300 | Baolin Sun |
| Strategic priority research program of Chinese Academy of Science | XDB29020000 | Baolin Sun |

## AUTHOR CONTRIBUTIONS

Yi Huang, Conceptualization, Data curation, Formal analysis, Investigation, Methodology, Project administration, Writing – original draft, Writing – review and editing | Md Roushan Ali, Investigation, Writing – review and editing | Wei Li, Investigation, Resources, Writing – review and editing | Wanying Wang, Resources | Yuanyuan Dai, Resources | Huaiwei Lu, Resources | Zhien He, Data curation, Investigation, Methodology, Writing – review and editing | Yujie Li, Funding acquisition, Resources | Baolin Sun, Funding acquisition

## DATA AVAILABILITY

Whole-genome sequencing data were deposited in the NCBI database and are publicly available in BioProject (accession numbers PRJNA956430 and PRJNA917695).

## ETHICS APPROVAL

Ethical review and approval were not required for the study on human participants in accordance with the local legislation and institutional requirements. Written informed consent for participation was not required for this study in accordance with the national legislation and the institutional requirements.

## ADDITIONAL FILES

The following material is available online.

### Supplemental Material

**Supplemental figures (mSystems00731-23-s0001.docx).** Figures S1 to S4.
**Supplemental tables (mSystems00731-23-s0002.xlsx).** Tables S1 to S3.

### Open Peer Review

**PEER REVIEW HISTORY (review-history.pdf).** An accounting of the reviewer comments and feedback.

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
