## [Reviewer comments · mSystems]

Epidemiological characteristics of multidrug-resistant *Acinetobacter baumannii* ST369 in Anhui, China

Yi Huang, Wei Li, Wanying Wang, Md Roushan Ali, Yuanyuan Dai, Huaiwei Lu, Yujie Li, Zhien He, and Baolin Sun

Corresponding Author(s): Baolin Sun, University of Science and Technology of China

Review Timeline:

Submission Date:

July 17, 2023

Accepted:

July 19, 2023

Editor: Sergio Baranzini

Reviewer(s): The reviewers have opted to remain anonymous.

Transaction Report:

DOI: <https://doi.org/10.1128/mSystems.00731-23>

July 19, 2023

Prof. Baolin Sun
University of Science and Technology of China
Department of Oncology
Huangshan Road
Hefei, Anhui 230027
China

Re: mSystems00731-23 (Epidemiological characteristics of multidrug-resistant *Acinetobacter baumannii* ST369 in Anhui, China)

Dear Prof. Baolin Sun:

Your manuscript has been accepted, and I am forwarding it to the ASM Journals Department for publication. For your reference, ASM Journals' address is given below. Before it can be scheduled for publication, your manuscript will be checked by the mSystems production staff to make sure that all elements meet the technical requirements for publication. They will contact you if anything needs to be revised before copyediting and production can begin. Otherwise, you will be notified when your proofs are ready to be viewed.

If you would like to submit a potential Featured Image, please email a file and a short legend to msystems@asmusa.org. Please note that we can only consider images that (i) the authors created or own and (ii) have not been previously published. By submitting, you agree that the image can be used under the same terms as the published article. File requirements: square dimensions (4" x 4"), 300 dpi resolution, RGB colorspace, TIF file format.

We recognize that the video files can become quite large, and so to avoid quality loss ASM suggests sending the video file via <https://www.wetransfer.com/>. When you have a final version of the video and the still ready to share, please send it to mSystems staff at msystems@asmusa.org.

Sincerely,

Sergio Baranzini
Editor, mSystems

Journals Department
E-mail: mSystems@asmusa.org